🔓 | Open Peer Review | Bacteriophages | Research Article

# Dysbiosis of fecal virome in pediatric Crohn's disease and its dynamic changes during infliximab therapy

Ting Ge,[1,2] Tingting Zhao,[3,4] Yangming Ruan,[1,2] Lin Ye,[1,2] Yongmei Xiao,[1] Fangfei Xiao,[1,2] Youran Li,[1,2] Xiaolu Li,[1,2] Ruixue Wang,[1,2] Hui Hu,[1] Chunyan Lu,[1] Hong Sun,[3,4] Chiyu Zhang,[5] Guangjun Yu,[4,6] Ting Zhang[1,2,4]

**ABSTRACT** The gut virome is an emerging but underexplored component of the human microbiota, especially in pediatric Crohn's disease (CD). This study aimed to characterize the fecal virome in children with CD and evaluate its association with clinical response to infliximab (IFX) therapy. A total of 85 participants, including 60 pediatric CD patients and 25 healthy controls (HC), were recruited. Among the CD patients, 53 received ≥3 IFX infusions, 41 achieved remission (IFX-R), and 12 did not (IFX-NR). Viral-like particles in fecal samples were enriched and profiled by metagenomic sequencing, while bacterial communities were assessed via 16S rRNA gene sequencing. Pediatric CD patients exhibited significantly reduced viral richness and altered viral community compared to HCs. Functional analyses revealed that CD patients exhibit a shift in fecal virome function from DNA repair to viral replication and assembly. Trans-kingdom correlations were disrupted in CD, particularly between *Torque teno* viruses and beneficial bacteria, such as *Blautia*. An integrated machine learning model combining viral and bacterial markers achieved a certain level of diagnostic accuracy for pediatric CD (area under the curve [AUC] = 89.3%). IFX treatment influences the gut virome, with remission associated with higher abundances of *Microviridae* and *Siphoviridae*, while *Anelloviridae*, *Myoviridae,* and *Podoviridae* were enriched in IFX-NR at baseline. These findings suggest the virome as a potential biomarker for predicting clinical outcome in pediatric CD, offering a novel avenue for disease diagnosis and personalized treatment strategies.

**IMPORTANCE** Crohn's disease (CD) in children poses a growing clinical challenge, with increasing incidence and variable response to biologic therapies such as infliximab (IFX). While gut bacterial dysbiosis has been extensively studied, the role of the gut virome in pediatric CD remains largely unexplored. This study provides the first longitudinal characterization of the fecal virome in children with CD undergoing IFX therapy. We reveal distinct viral community patterns, functional alterations, and virus-bacteria interactions in pediatric CD patients. Notably, integration of virome and bacteriome profiles enhances diagnostic accuracy, offering a promising avenue for predictive biomarker development. Furthermore, virome changes may be associated with the IFX treatment outcomes in children with CD. These findings highlight the gut virome as a critical but overlooked dimension of host-microbiome interactions in pediatric CD, with potential implications for personalized therapy and mechanistic understanding of treatment resistance.

**KEYWORDS** Crohn's disease, fecal virome, gut microbiome, infliximab, pediatric

Crohn's disease (CD) is a prominent subtype of inflammatory bowel disease (IBD) distinguished by chronic transmural inflammation, mucosal aphthous ulcers, and noncaseating granulomas in the gastrointestinal (GI) tract (1). Over the past two decades,

**Peer Reviewers** Qiyi Chen, Shanghai Tenth People's Hospital, Shanghai, China; Sahar M. Jawad Abduladheem, University of Kufa, Al Najaf, Iraq

Address correspondence to Guangjun Yu, gjyu@shchildren.com.cn, or Ting Zhang, zhangt@shchildren.com.cn.

Ting Ge and Tingting Zhao contributed equally to this article. Author order was determined both alphabetically and in order of increasing seniority

The authors declare no conflict of interest.

See the funding table on p. 15.

the incidence of pediatric-onset IBD has been on the rise globally, including in China (2, 3). Notably, the rising prevalence of CD among children and adolescents has rendered it an increasingly common pediatric GI condition (4). The primary clinical features of CD include diarrhea, abdominal pain, fever, fistula formation, and perianal lesions. The disease can affect the entire GI and is often accompanied by systemic manifestations (1). Due to its chronic, progressive, and relapsing course, CD can profoundly impair growth, development, and psychological health in pediatric patients, while also placing a considerable emotional and financial burden on their families (3).

Although the exact etiology and pathogenesis of CD remain incompletely understood, emerging evidence highlights the critical role of the gut microbiota (5–7). Within the context of genetic susceptibility, environmental exposures, and dietary factors, microbial dysbiosis may disrupt intestinal immune homeostasis, triggering an exaggerated immune response that ultimately leads to chronic intestinal inflammation (3, 8). In recent years, increasing attention has been directed toward another critical yet understudied component of the gut microbiota, the gut virome. Comprising predominantly bacteriophages and eukaryotic viruses, the virome represents a dynamic and integral part of the intestinal ecosystem, with emerging evidence suggesting its potential role in shaping host immunity and influencing disease outcomes (9, 10). The gut virome plays a pivotal role in modulating bacterial community structure and function. By influencing microbial dynamics and interacting with the host immune system, it may contribute to the initiation and progression of IBD (11–13).

At present, there is no definitive cure for CD, and clinical management remains primarily focused on symptom control and long-term maintenance, with the goal of achieving mucosal healing. Current therapeutic strategies include aminosalicylates, immunomodulators, corticosteroids, biologic agents, and surgical interventions. However, these treatments are often limited by variable efficacy, high relapse rates, and the potential for significant adverse effects (14). The advent of anti-tumor necrosis factor (TNF) monoclonal antibodies and the emergence of novel biologic agents have ushered in a new era in the management of CD. While biologics have substantially enhanced initial clinical response rates, a significant subset of patients fails to achieve adequate therapeutic response. Furthermore, among those who initially respond, secondary loss of response over time remains a common and challenging clinical issue (15). Failure to achieve mucosal healing is associated with a high risk of disease relapse, progressive disability, and increased healthcare costs (3, 16). Improving the clinical response to biologic therapies remains a major challenge in the management of CD. Recent studies, including our preliminary investigations, have examined the associations between key microbial components (bacteria and fungi) and treatment outcomes in CD. These findings suggest that specific microbial signatures and their metabolites are closely linked to therapeutic response, particularly to anti-TNF agents (17–20). However, the relationship between the gut virome and the onset, progression, and clinical outcomes of CD remains largely unexplored, especially with respect to pediatric CD patients undergoing biologic treatment.

In this study, we aimed to characterize the fecal virome composition and explore its associations with the clinical response to infliximab (IFX) treatment in a cohort of pediatric CD patients.

## MATERIALS AND METHODS

### Study population

The study was approved by the Regional Ethics Committee of Shanghai Children's Hospital. Sixty pediatric CD patients and 25 healthy controls (HC) were prospectively enrolled in this study. Among the CD patients, 53 received at least three doses of IFX treatments. Inclusion criteria were newly diagnosed CD children with ages ranging from 6 to 16 years, excluding any patients with very-early-onset IBD. The diagnosis of CD was established according to the Porto criteria, and active disease was defined

using the Pediatric Crohn's Disease Activity Index (PCDAI) (21). Clinical monitoring included documentation of infections, medication use (particularly antibiotics), and other relevant health events. Peripheral blood was collected to measure white blood cells (WBC), platelets (PLT), hemoglobin (Hb), hematocrit, C-reactive protein (CRP), erythrocyte sedimentation rate (ESR), and albumin. IFX was administered intravenously at 5 mg/kg at weeks 0, 2, and 6, followed by maintenance infusions every 8 weeks (22). All 53 IFX-treated patients maintained therapeutic serum drug levels (3–7 µg/mL) during the study. CD patients were classified based on their clinical response to IFX treatment. Remission was defined as a post-treatment PCDAI score ≤10, while patients with PCDAI >10 were categorized as non-remission. Accordingly, patients were divided into the remission group (IFX-R, $n = 41$) and the non-remission group (IFX-NR, $n = 12$). HC was recruited from Shanghai Children's Hospital during routine check-ups. They had no history of chronic illness and were free of GI or allergic symptoms at the time of stool collection. Individuals who had used antibiotics within 3 months or presented with any inflammatory conditions were excluded.

## Study samples

Fecal samples for virome and bacteriome analysis were collected from all participants. A total of 81 stool samples were obtained from CD patients, including 45 at baseline and 36 after the third IFX infusion. Each HC subject provided one fecal sample, yielding 25 samples. For bacteriome analysis, 73 samples were processed, including 37 at baseline and 36 after the third IFX infusion, together with 24 samples from HC subjects. All specimens were immediately stored at −80°C until further processing.

## Enrichment and sequencing of virus-like particles from fecal samples

Fecal samples were processed and enriched for virus-like particles (VLPs) following previously described protocols with slight modifications (23). Briefly, approximately 300–400 mg of stool was suspended, filtered, and treated with lysozyme and DNase/RNase to eliminate non-viral nucleic acids. Total nucleic acids (including DNA and RNA) were then extracted using the Magnetic Viral DNA/RNA Kit (TIANGEN), amplified with the REPLI-g WTA Single Cell Kit (QIAGEN) (24, 25), and subjected to high-throughput sequencing on an Illumina NovaSeq X Plus platform.

## Viral sequencing data quality control and analysis

Raw reads were first processed with fastp (v0.24.0, -l 75) to filter reads with low quality. Host-derived reads were then excluded by aligning the filtered reads to the human reference genome (hg38) using Bowtie2 (v2.5.4, parameters: --end-to-end --sensitive). SortMeRNA (v4.3.7, --paired_in –fastx) was used to filter rRNA with the default database. All quality-filtered reads were retained for downstream analysis. Taxonomic profiling of the quality-filtered reads was conducted using Phanta, a tool specifically designed for human gut microbiome analysis. To account for sequencing depth, the bacterial and viral coverage thresholds were set to 0.01 and 0.05, respectively, as recommended. A species was considered present in the sample only if at least 10 reads were confidently classified to it. Viral abundance across taxonomic levels was determined by excluding reads assigned to bacterial and archaeal kingdoms. The alpha diversity was measured using the Chao1, Shannon, and Simpson indexes. The beta diversity among samples was calculated through principal coordinate analysis (PCoA) to Bray-Curtis distance.

## Virome function analysis

The quality-filtered reads were assembled using Megahit (v1.2.9) using default parameters with a minimum contig length of 1,000 bp. The contigs were first blasted against the nt_viruses database (https://ftp.ncbi.nlm.nih.gov/blast/db/, 2025.3.17) using Blast (v2.16.0 +). Contigs meeting the following criteria were classified as candidate viral contigs: alignment similarity ≥ 80%, the alignment length ≥ 500 bp, and $e \leq$ 1e−5.

To minimize false positives, these candidate viral contigs were further aligned against bacterial and archaea databases (extracted from the default Kraken2 database using the same thresholds). Contigs showing significant matches to non-viral sequences were discarded. All quality-filtered reads were mapped to the remaining viral contigs using Bowtie2 to identify virus-derived reads for functional annotation.

Functional annotation of virus sequences was performed using HUMAnN 3.0 based on gene ontology (GO) family databases. Predictive functions were collapsed by gene family identity, and abundance values were expressed as reads per kilobase (RPK). To facilitate comparisons between samples with different sequencing depths, CPM (copies per million) units are implemented to normalize HUMAnN's default RPK values prior to performing statistical analyses.

## 16S rRNA sequencing and analysis

Genomic DNA was extracted from fecal samples using the QIAamp DNA Stool Mini Kit (Qiagen, Germany) combined with bead-beating, as previously described. DNA concentration was quantified with Qubit 3.0 fluorometer and normalized to 50 ng/µL for downstream 16S rRNA gene sequencing. Sequencing libraries were prepared with the TruSeq DNA PCR-Free Kit (Illumina) and sequenced on an Illumina NovaSeq 6000 platform. Paired-end reads were first assigned to samples using unique barcodes and trimmed to remove barcode and primer sequences. The reads were then merged using FLASH (v1.2.11) to generate raw tags, followed by quality filtering with fastp (v0.23.1) to obtain high-quality clean tags. Chimera sequences were detected and removed using the vsearch package (v2.16.0) against the Silva (16S/18S), yielding effective tags. DADA2 pipeline implemented in QIIME2. Briefly, the DADA2 plugin in the QIIME2 software was used to generate amplicon sequencing variants (ASVs) of individual samples and construct an ASVs abundance table. Taxonomical assignment of the preprocessed ASVs was performed using the QIIME2 classify-sklearn method, and the reference database used for annotation was Silva 138.1. To obtain the 16S relative microbiome profiling matrix, the absolute abundance of ASVs was normalized using a standard of sequence number corresponding to the sample with the least sequences. The alpha diversity was measured using the Chao1, Shannon, and Simpson indexes. The beta diversity among samples was calculated through PCoA to Bray-Curtis distance based on the ASV abundance.

## Statistical analysis

All statistical analyses were performed using R software. Statistical differences were assessed by the Wilcoxon rank-sum test for comparison between two groups with $P$ values adjusted for Benjamini-Hochberg. Spearman's rank correlation was applied to assess relationships between variables. Linear discriminant analysis effect size (LEfSe) was performed to identify metagenomic biomarkers. Moreover, XGBoost was used to construct a classification model evaluating the diagnostic potential of the fecal virome and bacteriome, with 80% of samples for training and 20% for testing. Hyperparameters were tuned via fivefold cross-validation.

## RESULTS

### Clinical characteristics and outcomes of the pediatric CD

A total of 60 pediatric patients with CD and 25 sex- and age-matched HC subjects were enrolled in the study. Detailed clinical characteristics are summarized in Table 1. The median age of CD patients was 12.8 years (IQR: 11.4–13.8), and 36 were male. CD patients had significantly lower body mass index (BMI) compared to HC subjects, and most presented with ileocolonic involvement (L3). Among the CD cohort, 53 patients received at least three doses of IFX infusions, of whom 41 achieved clinical remission (IFX-R) and 12 did not (IFX-NR). At baseline, there were no significant differences in laboratory and clinical parameters, PCDAI, and fecal calprotectin (FCP) between the remission group and

**TABLE 1** Clinical characteristics of pediatric CD patients and HC[a]

| Characteristic | HC (n = 25) | CD (n = 60) | P value |
|---|---|---|---|
| Age, years | 12.3 (12.0, 12.9) | 12.8 (11.4, 13.8) | 0.426 |
| Male, n (%) | 13 (52.0) | 36 (60.0) | 0.496 |
| BMI, kg/m² | 18.7 (16.9, 20.4) | 14.9 (13.8, 17.0) | 0.001 |
| Location | | | |
| L1, n (%) | | 4 (6.7) | |
| L2, n (%) | | 3 (5.0) | |
| L3, n (%) | | 50 (83.3) | |
| L4, n (%) | | 3 (5.0) | |
| Clinical index | | | |
| WBC (×10⁹/L) | | 9.3 (6.9, 11.9) | |
| PLT (×10⁹/L) | | 441 (329, 539) | |
| CRP (mg/L) | | 31 (7, 51) | |
| ESR (mm/h) | | 82 (46, 109) | |
| PCDAI | | 32.5 (20, 42.5) | |
| FCP (μg/g) | | 450.0 (176.3, 450) | |
| Outcome (n = 53) | | | |
| Remission | | 41 (77.4) | |
| Non-remission | | 12 (22.6) | |

[a]The data were analyzed using the nonparametric Wilcoxon rank-sum test (two groups).

non-remission group (Table S1). In the remission group, significant improvements were observed in WBC, PLT, CRP, ESR, Hb, PCDAI, and FCP following IFX treatment. In contrast, only ESR showed a significant decrease after IFX treatment in the non-remission group (Table 2).

## Diversity and compositional alterations of fecal virome in pediatric CD

A total of 27,064,734 high-quality reads per sample were obtained from VLPs metagenomic sequencing and subsequently used for taxonomic profiling with Phanta (26). At both the class and family levels, the fecal virome displayed a high degree of individual variability, with particularly pronounced heterogeneity among CD patients (Fig. S1A and B). The average relative abundance of the major virome at the class level and the top 10 at the family level between pediatric CD and HC is shown in Fig. S1C and D, highlighting significant differences in virome composition. The predominant abundance in HC was *Malgrandaviricetes*, *Caudoviricetes*, and *Faserviricetes*, whereas *Caudoviricetes* and *Malgrandaviricetes* were more prevalent in the pediatric CD patients (Fig. S1C). No significant differences in the alpha diversity of virome communities were observed between CD patients and HC, with the exception of Chao 1 index (Fig. 1A). To assess overall virome composition, beta diversity was evaluated using PCoA based on Bray-Curtis dissimilarity. PERMANOVA testing confirmed a modest but statistically significant difference in viral community structure between the two groups ($R^2 = 0.02$, $P = 0.002$, Fig. 1B). LEfSe identified specific viral signatures associated with disease status. Pediatric CD patients showed significantly higher linear discriminant analysis (LDA) scores for *Caudoviricete* and *Anelloviridae* compared to HC. In contrast, *CrAss-like viruses*, *Microviridae*, and *Inoviridae* had higher LDA scores in HC (Fig. 1C). Inter-group comparisons of taxonomic profiles at the family level showed that fecal samples from pediatric CD patients had higher relative abundances of *Anelloviridae*, *Genomoviridae*, *Myoviridae*, and *Siphoviridae*, while *Inoviridae*, *Microviridae*, *Podoviridae*, and *Virgaviridae* were less abundant compared to HC. Statistically significant differences were observed for *Anelloviridae*, *Microviridae*, and *Virgaviridae* (Fig. 1D). Furthermore, a higher proportion of temperate phages and a lower proportion of virulent phages were observed in the CD fecal virome compared to HC (Fig. S2).

**TABLE 2** Changes in clinical and laboratory parameters before and after IFX treatment in remission and non-remission groups[a]

| Parameter | Remission ($n = 41$) | | P value | Non-remission ($n = 12$) | | P value |
|---|---|---|---|---|---|---|
| | Baseline | Third IFX | | Baseline | Third IFX | |
| Age, years | 12.9 (11.9, 13.9) | | | 12.0 (10.3, 13.7) | | 0.275 |
| Male, n (%) | 27 (66.8) | | | 3 (25.0) | | 0.012 |
| BMI, kg/m$^2$ | 15.1 (13.9, 16.9) | | | 14.0 (12.8, 15.3) | | 0.215 |
| Location | | | | | | |
| L1, n (%) | 3 (7.3) | | | 0 (0) | | |
| L2, n (%) | 1 (2.4) | | | 1 (8.3) | | |
| L3, n (%) | 35 (85.4) | | | 10 (83.4) | | |
| L4, n (%) | 2 (4.9) | | | 1 (8.3) | | |
| Clinical index | | | | | | |
| WBC (×10$^9$/L) | 9.6 (7.4, 11.6) | 5.8 (4.6, 7.1) | 0.001 | 8.2 (6.9, 12.8) | 5.1 (4.8, 8.3) | 0.014 |
| PLT (×10$^9$/L) | 475.0 (363.0, 567.0) | 288 (249.5, 340.0) | 0.001 | 435.0 (331.3, 546.5) | 349 (291.2, 397.2) | 0.414 |
| CRP (mg/L) | 32.0 (14.5, 49.0) | 5.0 (5.0, 5.0) | 0.001 | 37.5 (7.5, 72.5) | 5.0 (5.0, 5.0) | 0.102 |
| ESR (mm/h) | 86.0 (47.0, 108) | 15.0 (8.0, 29.0) | 0.001 | 88.5 (44.8, 120.0) | 28.0 (14.2, 96.0) | 0.014 |
| Hb (g/L) | 106.0 (98.5, 118.5) | 127.0 (120.0, 135.0) | 0.001 | 98.5 (86.5, 20) | 117.0 (108.0, 138.0) | 0.146 |
| PCDAI | 32.5 (26.0, 42.5) | 5.0 (5.0, 5.0) | 0.000 | 31.3 (13.0, 45.9) | 12.3 (10.0, 24.6) | 0.102 |
| FCP (µg/g) | 450.0 (162.2, 450.0) | 121.0 (47.3, 310.4) | 0.025 | 419.6 (120.3, 817.4) | 341.1 (70.5, 550.1) | 0.558 |

[a]The data were analyzed using the nonparametric Wilcoxon rank-sum test (two groups).

## Alterations in fecal virome function in pediatric CD

To investigate functional changes in the fecal virome, viral sequences were annotated using HUMAnN3 with the GO database, and differentially enriched functions were identified via LEfSe analysis. Compared to HC, CD patients exhibited significant shifts in viral functional profiles. In the HC, functions such as DNA helicase activity and DNA-dependent ATPase activity were enriched. In contrast, CD patients showed enrichment of viral functions related to genome replication (e.g., RNA helicase activity), capsid formation ($T = 1$ icosahedral structure), and energy metabolism (uncoupled ATPase activity; Fig. 2).

## Diversity and compositional alterations of fecal bacteriome in pediatric CD

Bacterial 16S rRNA gene sequencing was performed to examine alterations in the gut bacteriome of pediatric CD patients relative to HC. Both bacterial richness and diversity were notably reduced in CD patients (Fig. 3A through C). Beta diversity analysis showed distinct clustering of samples based on CD status ($R^2 = 0.11$, $P = 0.001$; Fig. 3D). At the family level, inter-group comparisons of taxonomic profiles revealed that fecal samples from pediatric CD patients had lower relative abundances of *Acidaminococcaceae*, *Bacteroidaceae*, *Butyricicoccaceae*, *Christensenellaceae*, *Desulfovibrionaceae*, *Lachnospiraceae*, *Marinifilaceae*, *Monoglobaceae*, *Oscillospiraceae*, *Peptostreptococcaceae*, *Rikenellaceae*, *Ruminococcaceae*, and *Tannerellaceae*, but higher relative abundances of *Aerococcaceae*, *Enterococcaceae*, and *Morganellaceae* compared to HC (Fig. 3E).

## Trans-kingdom correlations between fecal virome and bacteriome in CD

Gut viral-bacterial trans-kingdom correlations play important roles in health and disease (27, 28). We further explored specific trans-kingdom associations between bacterial and viral species that significantly differed in CD and HC. In particular, *Torque teno viruses 13* and *Escherichia virus phiV10* displayed significant negative correlations with several beneficial bacterial species, including *Blautia obeum* and *Romboutsia* (Fig. 4).

## Integration of fecal virome and bacteriome enhances diagnostic accuracy for pediatric CD

To assess the diagnostic potential of the fecal virome, we developed an XGBoost-based machine learning model to distinguish pediatric CD patients from HC. A viral classifier

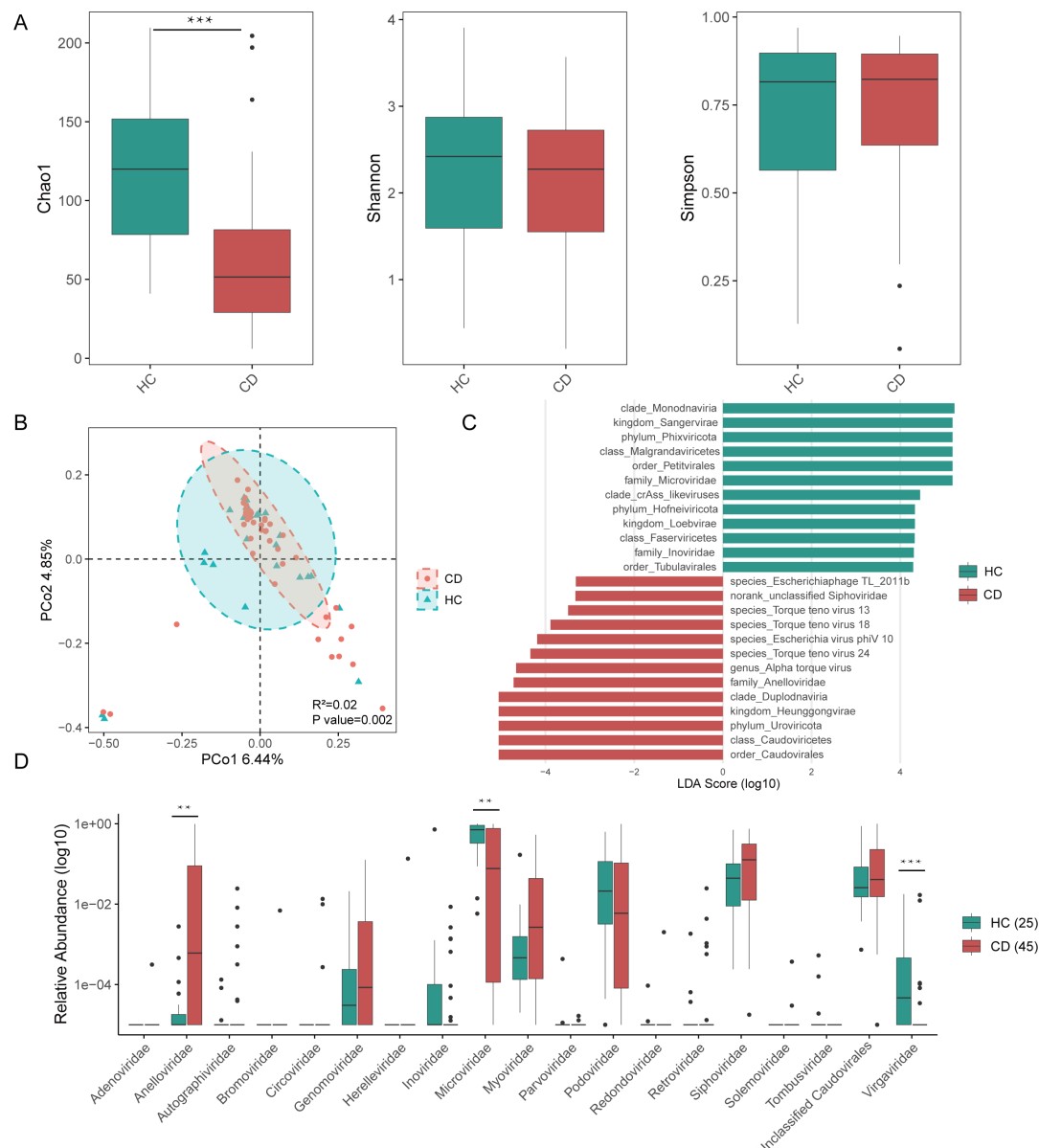

**FIG 1** Altered fecal virome diversity and composition in pediatric CD. (A) Alpha diversity of the fecal virome in CD patients compared to HCs, assessed by the Chao1 index, Shannon index, and Simpson index. (B) Beta diversity analysis based on Bray-Curtis dissimilarity, visualized by PCoA to compare overall virome composition between groups. (C) LDA scores of differentially abundant viral taxa between CD patients and HC, as identified by LEfSe ($P < 0.05$, LDA > 3). (D) Boxplots showing the significantly different virus family level. Significance is determined by using the Wilcoxon rank-sum test, with $*P < 0.05$, $**P < 0.01$, and $***P < 0.001$.

incorporating five key species achieved an area under the curve (AUC) of 85.7% (Fig. 5A). Similarly, a model based solely on bacterial features yielded comparable performance (AUC = 85.7%; Fig. 5B). Notably, combining virome and bacteriome features resulted in improved diagnostic performance, with the integrated model achieving an AUC of 89.3% (Fig. 5C and D), outperforming models using either data type alone.

## IFX modified the fecal virome

To investigate the dynamic changes of gut virome during IFX therapy, we compared the virome communities in pre-treatment (baseline) and post-treatment (post-IFX) fecal samples. To reduce the effect of inter-individual variability, only pediatric CD patients

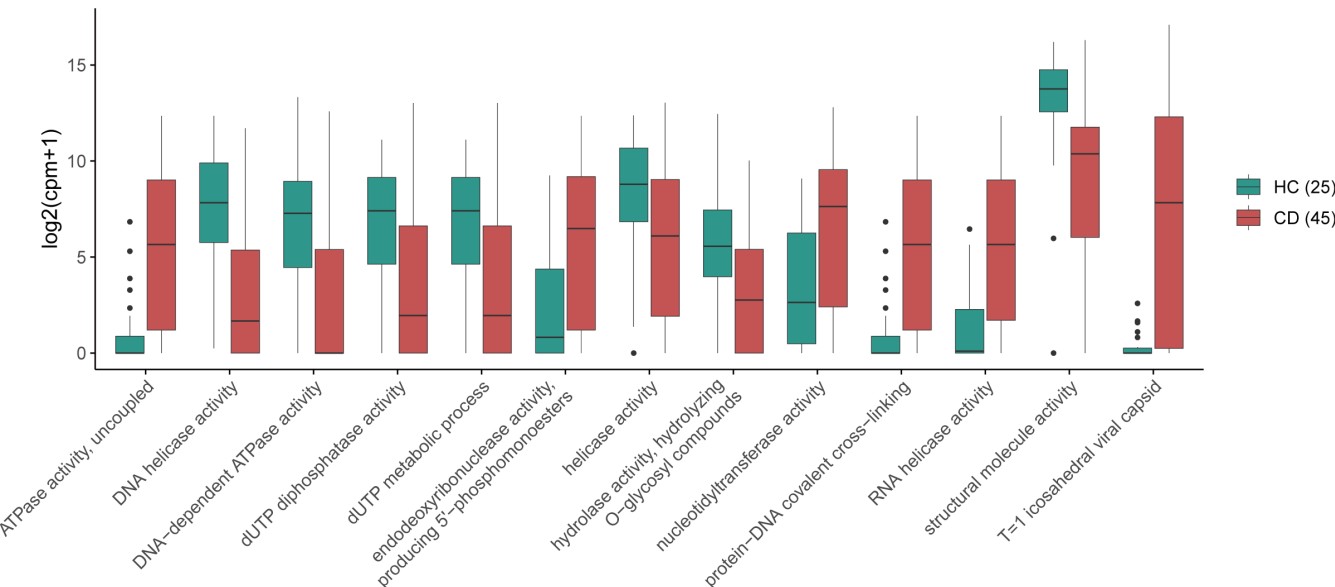

**FIG 2** Functional profiling of the fecal virome based on GO. Boxplots depict the relative abundance (log10-transformed) of key viral functions, where these functions were identified by LEfSe ($P < 0.05$, LDA > 3).

($n$ = 25) who provided both baseline and post-treatment samples were included in the analysis. Virome community diversity did not show significant changes before and after IFX treatment in CD patients, with the exception of Chao1 (Fig. 6A). PCoA analysis indicated that baseline and post-IFX samples formed distinct clusters, implying differences in microbial composition; however, this separation was not statistically significant ($R^2$ = 0.018, $P$ = 0.721; Fig. 6B). LEfSe analysis identified the viral taxa responsible for the observed differences: baseline samples were predominantly composed of *Pbunavirus* and *Pseudomonas virus E215*, while post-IFX samples exhibited a distinct range of genera and species (Fig. 6C). A noticeable change in the relative abundance of some families is evident between the two groups. The box plot shows the relative abundance of various viral families at baseline and post-IFX. Some families show clear differences in relative abundance between the two groups (e.g., *Anelloviridae* and *Microviridae*; Fig. 6D).

## Alterations of fecal virome associated with IFX treatment outcomes

We next tried to identify specific characteristics of the fecal virome that could serve as indicators of therapeutic response to IFX. Fecal samples from patients in remission were categorized as response samples, while those from non-remission patients were classified as non-response samples. Comparative analysis of taxonomic profiles from CD patients prior to IFX treatment revealed significantly higher relative abundances of several virome families in the non-remission group (Baseline-NR) compared to the remission group (Baseline-R), including *Anelloviridae*, *Myoviridae*, and *Podoviridae*. In contrast, the remission group (Baseline-R) exhibited higher levels of *Microviridae*, *Siphoviridae*, and *Unclassified Caudovirales* (Fig. 7A). After IFX treatment, remission patients (IFX-R) showed increased abundances of *Anelloviridae*, *Genomoviridae*, *Microviridae*, *Podoviridae*, and *Virgaviridae*, while non-remission patients (IFX-NR) displayed higher abundances of *Myoviridae*, *Siphoviridae*, and *unclassified Caudoviricetes* (Fig. 7B).

## DISCUSSION

Besides bacteria and fungi, viruses also make up a significant portion of the human gut microbiota. The viruses in the human gut include bacteriophages that infect prokaryotic

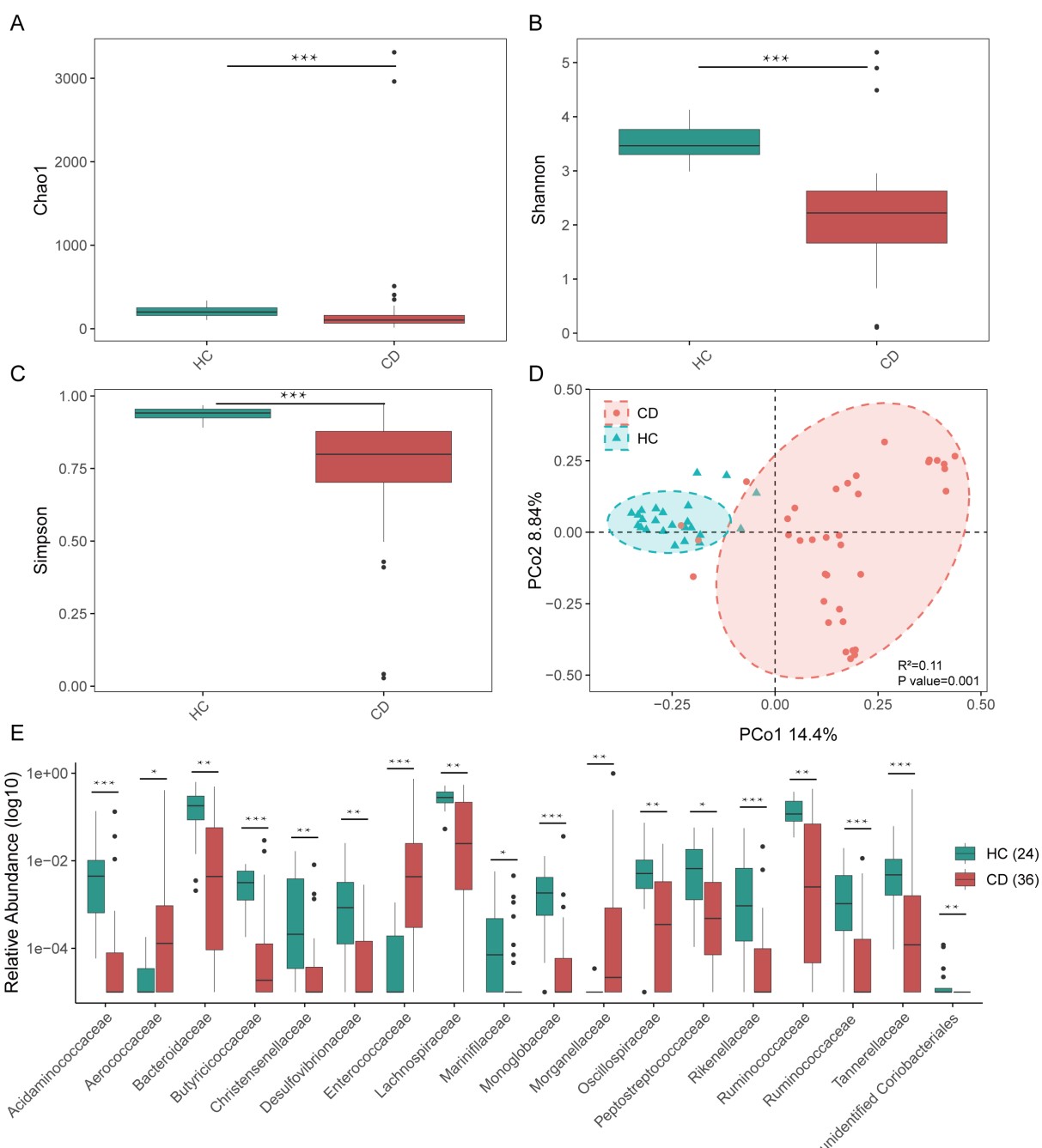

**FIG 3** Richness and diversity of the fecal bacteriome in pediatric CD. (A–C) Alpha diversity of the fecal bacteriome in CD patients compared to HCs, assessed by the Chao1 index, Shannon index, and Simpson index. (D) Beta diversity analysis based on Bray-Curtis dissimilarity, visualized by PCoA to compare overall bacteriome composition between groups. (E) Boxplots showing the significantly different bacterial family level. Significance is determined by using the Wilcoxon rank-sum test, with *$P < 0.05$, **$P < 0.01$, and ***$P < 0.001$.

cells, as well as DNA viruses and RNA viruses that infect eukaryotic cells, among which bacteriophages are the most predominant (11, 29). Most of these bacteriophages belong to the order *Caudovirales*, characterized by double-stranded DNA genomes. These bacteriophages are mainly represented by the families *Siphoviridae*, *Podoviridae*, *Myoviridae*, and *CrAss-like* phages (30). Dozens of different morphological combinations of these three families can be identified in fecal samples collected from different individuals (30). While the link between gut bacteria and CD is well established (9,

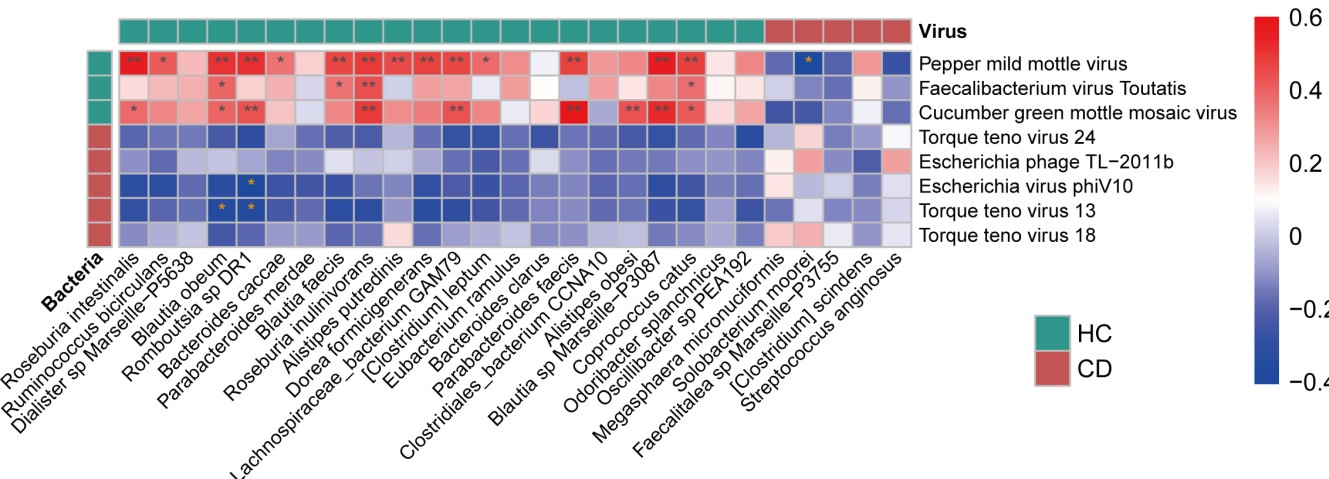

**FIG 4** Heatmap of the Spearman's correlation coefficients between differential DNA viral species and differential bacterial species in pediatric CD versus HCs, where these species were identified by LEfSe ($P < 0.05$, LDA > 3).

20, 31), the role of the fecal virome, especially in pediatric CD patients, remains largely unexplored.

Several previous studies have documented alterations in the gut virome of pediatric patients with IBD. Wagner et al. found that a metagenomics analysis of gut tissue and wash samples has characterized a large abundance of phages when compared with pediatric CD patients and control individuals (32). Fernandes et al. found that compared with healthy subjects, the abundance of *Microviridae* was reduced in CD patients (33). Liang et al. uncovered that compared with healthy subjects, very early-onset IBD consistently showed a marked increase in the relative abundance of *Caudovirales* and a reciprocal decrease in *Microviridae*, yielding a significantly elevated *Caudovirales/Microviridae* ratio that was further accentuated by immunosuppressive therapy (34). In parallel, the prevalence and load of *Anelloviridae* were significantly higher in pediatric IBD cohorts and correlated with immunosuppressive treatment (32, 34). Agreeing with previous findings (29, 31), we detected pronounced interindividual variability and heterogeneity in the fecal virome among pediatric CD patients, mirroring prior reports of the gut virome's intricate diversity. Our data revealed that pediatric CD patients showed significantly reduced fecal viral species richness (as measured by the Chao1 index) and changed virome communities as compared to HC subjects. Pediatric CD patients were enriched with families *Caudoviricete* and *Anelloviridae* with a high LDA score. Inter-group comparisons of taxonomic profiles revealed statistically significantly higher levels of *Anelloviridae* and lower levels of *Microviridae* and *Virgaviridae* in CD children than HCs. We also observed a higher proportion of temperate phages in the CD group, consistent with findings from previous studies (35). Temperate phages can modulate bacterial fitness and contribute to the functional and taxonomic diversity of the gut microbiota through lysogenic conversion and horizontal gene transfer, and they can deliver auxiliary metabolic genes that upregulate bacterial DNA repair, ATPase, and capsid synthesis (36). For example, in the intestinal mucosa of patients with ulcerative colitis (UC), numerous viral functional signatures, particularly those associated with phage-mediated enhancement of bacterial fitness and pathogenicity, are markedly upregulated (12). During our study, virome functions were also altered in pediatric CD patients, with an enrichment of viral functions related to genome replication, capsid formation, and energy metabolism, suggesting increased viral activity and replication. As reported (31), enriched virome functions in CD, such as viral DNA replication and eukaryotic viral capsid protein functions, were associated with virus‐host interaction.

Cross-kingdom interactions between gut viruses and bacteria are vital for preserving intestinal balance and influencing the course of disease (27, 28). Previous studies have

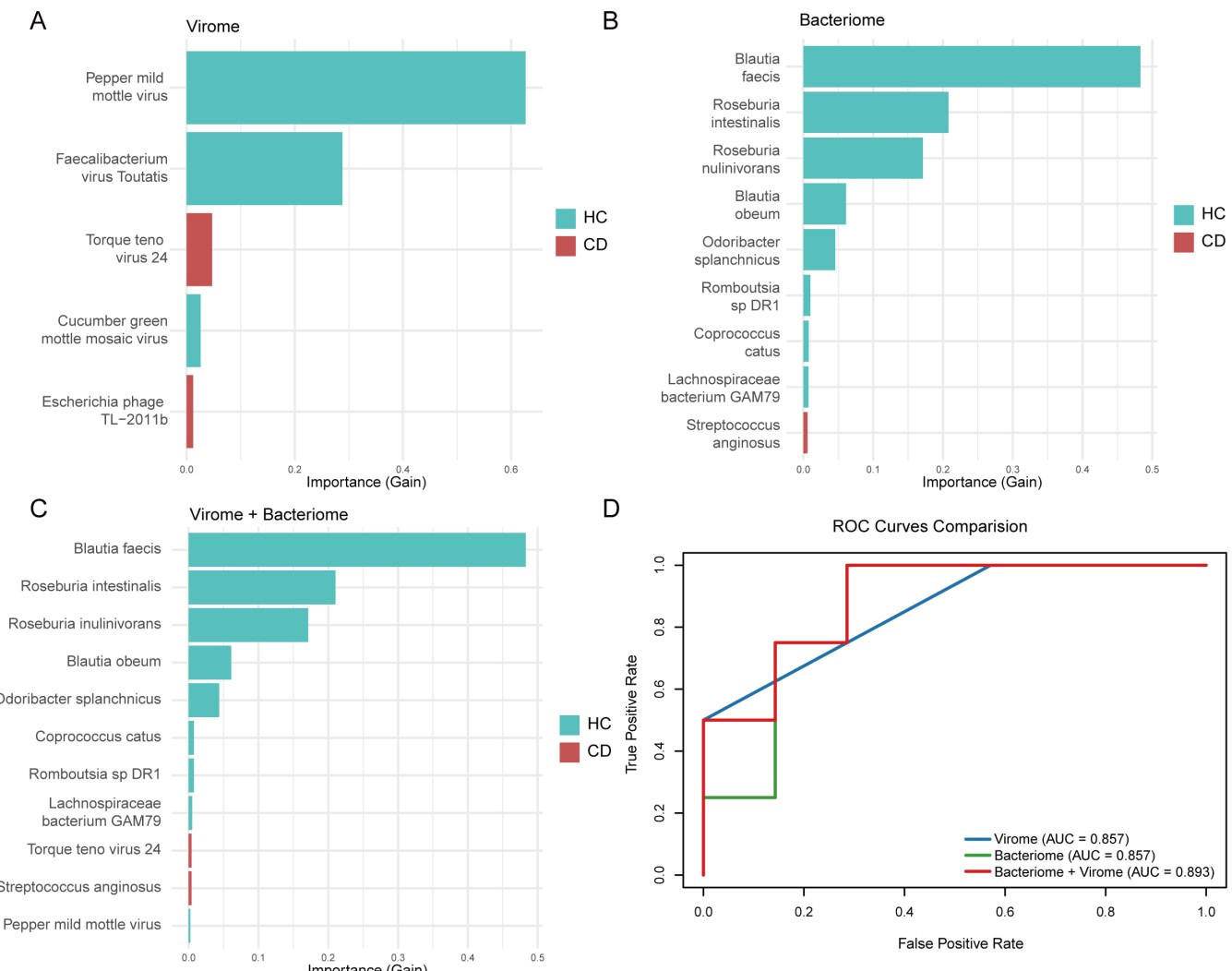

**FIG 5** Classification models construction between pediatric CD and HCs. (A) A fecal virome-based model. (B) A bacteriome-based model. (C) A combined virome and bacteriome model was constructed using XGBoost classifiers. (D) The receiver operating characteristic (ROC) curve analysis for each of the three models.

consistently reported reduced bacterial diversity, decreased abundance of butyrate-producing species such as *Faecalibacterium prausnitzii*, and an increased prevalence of adherent-invasive *Escherichia coli* in the gut microbiota of CD patients (9, 19, 20, 37). Bacteriophages, as crucial modulators of bacterial diversity and adaptability, are thought to play a significant role in shaping the structure and function of microbial communities (38). Significant alterations in virome-bacteriome associations were observed in CD patients, indicating a breakdown in the integrity of the gut microbial network (17). These changes highlight the potential role of the virome in preserving microbial equilibrium and supporting overall gut health. Previous work from our group characterized the gut bacteriome in pediatric CD (19, 20). Here, we extend this research by examining a larger cohort and integrating virome profiling. This approach uncovered a distinct reorganization of trans-kingdom associations in CD, characterized by the disappearance of certain bacterium-virus relationships and the appearance of new ones. The negative correlation between *T. teno* viruses and butyrate producers (*Blautia* and *Romboutsia*) suggests virus-associated suppression of anti-inflammatory bacteria, a pattern consistent with reported virome-bacteriome metabolic coupling in IBD (17). Recent research has highlighted the significance of interactions between gut phages and bacteria in influencing the protective effects of diet on CD (39). These alterations imply a potential

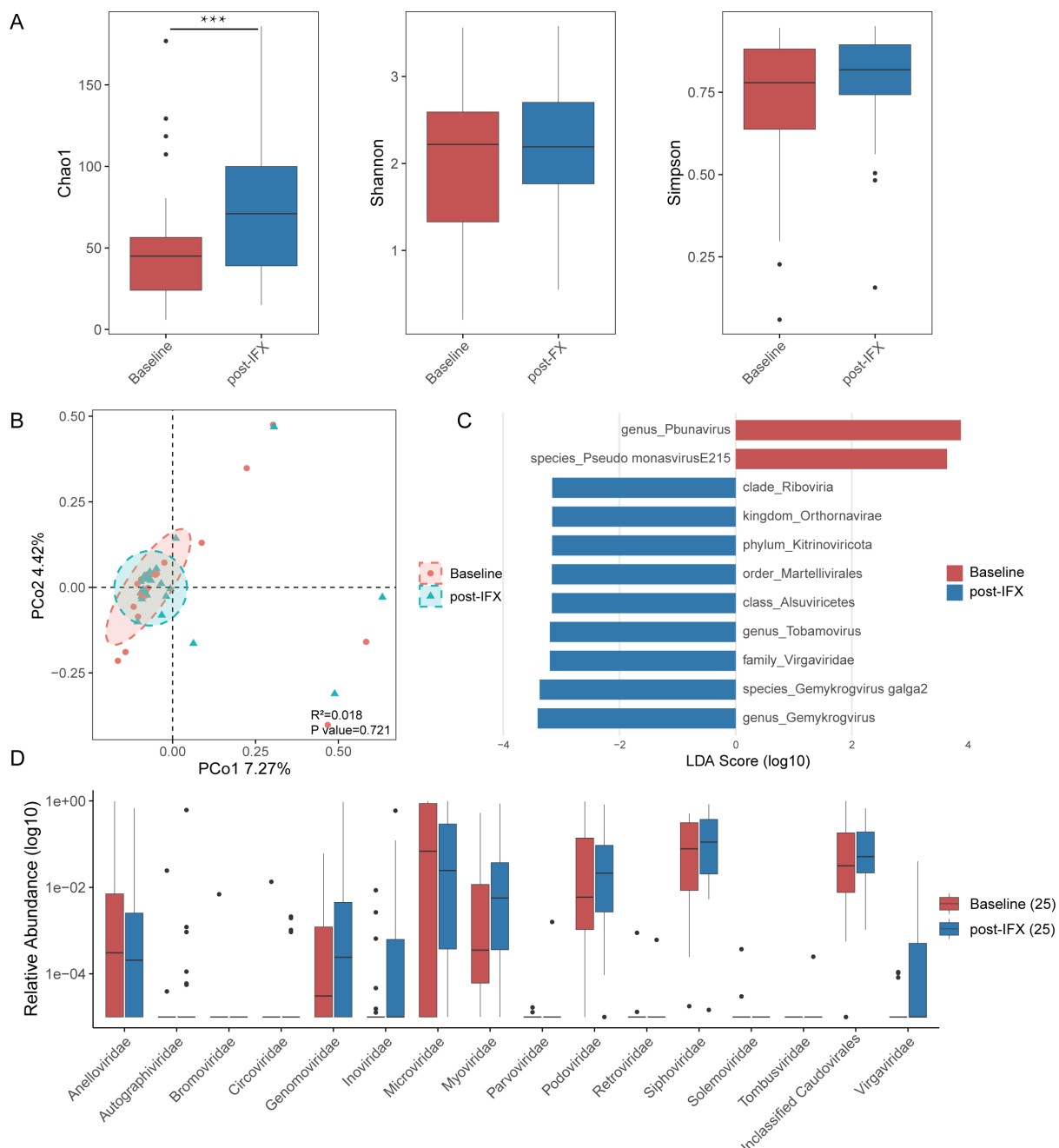

**FIG 6** Altered fecal virome diversity and composition in the pediatric CD patients after IFX treatment. (A) Alpha diversity of the fecal virome in the CD patients before and after IFX treatment was assessed by the Chao1 index, Shannon index, and Simpson index. (B) Beta diversity analysis based on Bray-Curtis dissimilarity, visualized by PCoA to compare overall virome composition between the CD patients before and after IFX treatment. (C) LDA scores of differentially abundant viral taxa between the CD patients before and after IFX treatment, as identified by LEfSe ($P < 0.05$, LDA > 3). (D) Boxplots showing the significantly different virus family level. Significance is determined by using the Wilcoxon rank-sum test, with *$P < 0.05$, **$P < 0.01$, and ***$P < 0.001$.

role of the gut virome, particularly bacteriophages, in modulating bacterial communities and contributing to disease pathogenesis. Given the complexity of host-microbiota interactions in humans, further mechanistic studies using animal models are warranted to elucidate the virome's contribution to the development of CD. We further developed classification models to differentiate CD patients from HC by using XGBoost. The results demonstrated that combining viral and bacterial features further improved classification accuracy compared to models based on either data set alone. These findings highlight

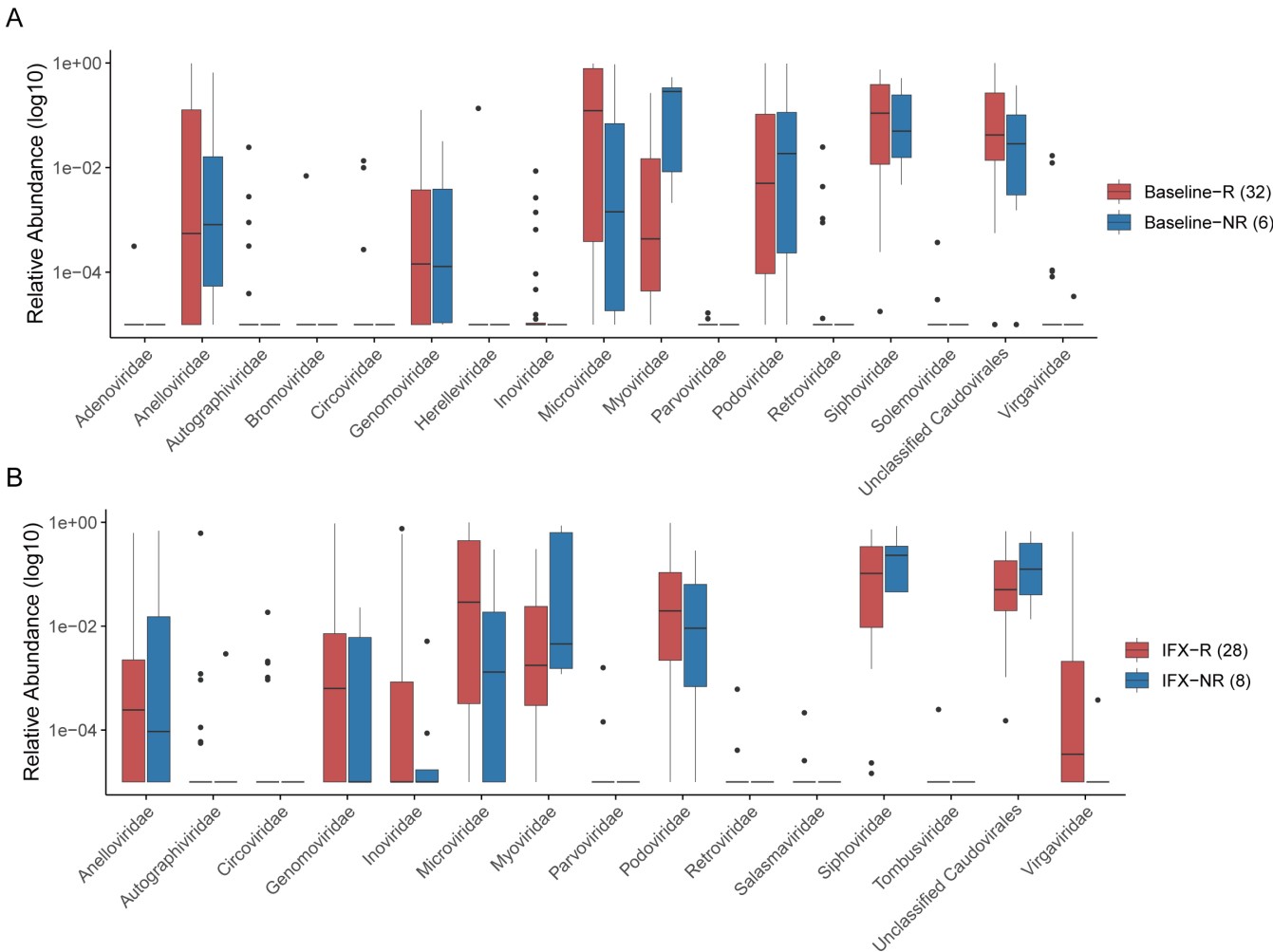

**FIG 7** Fecal viruses correlate with the outcome of the pediatric CD patients with IFX treatment. (A) Boxplots showing the significantly different virus family level between the remission (Baseline-R) and non-remission (Baseline-NR) patients before IFX treatment. (B) Boxplots showing the significantly different virus family level between the remission (IFX-R) and non-remission (IFX-NR) patients after IFX treatment. Significance is determined by using the Wilcoxon rank-sum test.

the limitations of single-omics approaches and emphasize the value of integrative multi-omics strategies, specifically the combination of virome and bacteriome data, for enhancing disease prediction in pediatric CD.

Anti-TNF therapy was first introduced for treating CD patients in 1992 and has become the preferred first-line treatment for moderate-to-severe cases that are refractory or dependent on corticosteroids. In clinical practice, the primary anti-TNF biologics used to treat pediatric CD are IFX and adalimumab. Jongsma et al. demonstrated that the clinical remission rate and endoscopic remission rate at week 10 were significantly higher in children treated with IFX compared to those receiving conventional therapy (40). Data from Bouhnik's study showed that by week 24, 64% of children with CD achieved clinical remission, and 45.7% of those maintained long-term remission over a 4-year follow-up. However, in real-world clinical practice, nearly 30% of CD patients fail to respond to initial anti-TNF therapy (41). Among those who respond to initial anti-TNF therapy, fewer than 50% are able to maintain a sustained clinical response over time (42). With the rapid development of multi-omics technologies, researchers have begun exploring the relationship between the gut microbiota and its metabolites and their influence on the response to anti-TNF therapy. At present, most studies examining the relationship between the gut microbiota and clinical response to anti-TNF

therapy in CD remain centered on bacterial components (19, 20, 43–46), with limited studies on viruses.

In this study, our results show that IFX treatment influences the gut virome in pediatric CD patients, particularly in terms of taxonomic composition and some diversity metrics. Our data showed that the baseline group was characterized by higher abundances of *Anelloviridae* and *Microviridae*, while *Genomoviridae*, *Myoviridae*, *Podoviridae*, *Siphoviridae*, and *Virgaviridae* were more abundant in the post-IFX group, aligning with earlier reports of virome disturbances in IBD (35, 47). These observed changes in the gut virome of children with CD highlight its important role in the development and progression of the condition. These observations reinforce the idea that virome dysbiosis is a hallmark of CD. Jansen et al. found that the gut virome segregates into two distinct clusters, designated CA and CrM, each correlating differently with response to anti-TNF therapy in adult IBD patients, and further suggested that these two gut virome configurations may be involved in the pathophysiology of IBD (48). To the best of our knowledge, this study is the first to explore the associations of bacteriophage profiles and anti-TNF treatment outcomes in pediatric CD. Our data showed that CD children who went on to achieve remission had significantly higher levels of *Microviridae*, *Siphoviridae*, and *unclassified Caudoviricetes* than the nonremission group, while the nonremission group showed an overrepresentation of *Myoviridae* and *Podoviridae* before initiating IFX therapy. Following IFX treatment, responders displayed higher viral diversity and a broader distribution of dominant taxa than non-responders, indicating a potential reshaping of the virome during disease remission.

Despite the novel insights of this study, our findings are preliminary and require validation in larger cohorts to confirm the observed trends. First, the small sample size limits statistical power and generalizability, particularly regarding the relationship between the fecal virome and IFX treatment outcomes in pediatric CD. Second, virome and bacteriome analyses were performed only on fecal samples collected at baseline and one post-treatment timepoint, which may dilute the time-dependent effect of IFX infusions and limit the ability to evaluate dynamic changes throughout the treatment course. Additionally, our sampling was restricted to fecal material, which may not fully represent mucosal virome communities more directly involved in CD pathogenesis. Third, unmeasured confounders, such as geographic factors, diet, and environmental exposure, may have influenced the microbiome composition (12, 49). Finally, this study focused on virome signatures rather than underlying mechanisms. Future research should adopt a longitudinal, multi-center design, incorporate mucosal tissue analyses, and use gnotobiotic animal models or *in vitro* systems to explore the functional roles of viruses in CD pathogenesis.

In summary, this study provides a comprehensive profile of gut virome alterations in pediatric CD, highlighting the pivotal role of the fecal virome in the CD pathogenesis. Furthermore, virome composition is closely associated with treatment outcomes following IFX therapy, which underscores the potential of virome-derived biomarkers as indicators of disease activity and predictors of IFX treatment response in pediatric patients with CD.

## ACKNOWLEDGMENTS

This work was supported by the grants from the National Natural Science Foundation of China (grant number 8247031679). The funders had no role in study design, data collection and analysis, decision to publish, or preparation of the manuscript.

Ting Zhang and G.Y. designed the study. T.G. and Tingting Zhao interpreted the data and wrote the manuscript. Ting Zhang analyzed the omics data and created figures. T.G., Y.R., L.Y., Y.X., F.X., Y.L., X.L., R.W., H.H., H.S., C.Z., and C.L. analyzed and interpreted the patient data regarding Crohn's disease. All authors read and approved the final manuscript. The authors have no conflicts of interest to disclose.

## AUTHOR AFFILIATIONS

[1]Department of Gastroenterology, Hepatology, and Nutrition, Shanghai Children's Hospital, School of Medicine, Shanghai Jiao Tong University, Shanghai, China

[2]Gut Microbiota and Metabolic Research Center, Institute of Pediatric Infection, Immunity and Critical Care Medicine, School of Medicine, Shanghai Jiao Tong University, Shanghai, China

[3]Center for Biomedical Informatics, Shanghai Children's Hospital, School of Medicine, Shanghai Jiao Tong University, Shanghai, China

[4]Shanghai Engineering Research Center for Big Data in Pediatric Precision Medicine, Shanghai, China

[5]Shanghai Public Health Clinical Center, Fudan University, Shanghai, China

[6]The Second Affiliated Hospital, School of Medicine, The Chinese University of Hong Kong, Shenzhen and Longgang District People's Hospital of Shenzhen, Shenzhen, China

## AUTHOR ORCIDs

Ting Ge http://orcid.org/0009-0004-2413-3294
Tingting Zhao http://orcid.org/0009-0006-7616-1654
Guangjun Yu http://orcid.org/0000-0002-3159-4652
Ting Zhang http://orcid.org/0000-0001-9391-8926

## FUNDING

| Funder | Grant(s) | Author(s) |
| --- | --- | --- |
| National Natural Science Foundation of China | 8247031679 | Ting Zhang |

## AUTHOR CONTRIBUTIONS

Ting Ge, Data curation, Formal analysis, Investigation, Methodology, Project administration, Resources, Software, Writing – original draft | Tingting Zhao, Methodology, Software, Validation, Visualization, Writing – original draft, Writing – review and editing | Yangming Ruan, Data curation | Lin Ye, Data curation, Methodology | Yongmei Xiao, Conceptualization, Formal analysis, Validation | Fangfei Xiao, Data curation, Methodology | Youran Li, Methodology | Xiaolu Li, Data curation, Formal analysis | Ruixue Wang, Data curation | Hui Hu, Data curation | Chunyan Lu, Data curation | Hong Sun, Methodology | Chiyu Zhang, Formal analysis, Methodology, Visualization | Guangjun Yu, Funding acquisition, Supervision, Writing – review and editing | Ting Zhang, Formal analysis, Funding acquisition, Methodology, Project administration, Resources, Supervision, Writing – review and editing

## DATA AVAILABILITY

Raw sequencing data are available in the NCBI BioProject database (https://www.ncbi.nlm.nih.gov/bioproject/) under study accession number PRJNA1282097.

## ETHICS APPROVAL

The ethics application (No.2024R130-E01) was approved by the Regional Ethical Review Board of the Shanghai Children's Hospital, School of Medicine, Shanghai Jiao Tong University.

## ADDITIONAL FILES

The following material is available online.

### Supplemental Material

**Supplemental file (mSystems01489-25-s0001.docx).** Table S1; Fig. S1 and S2.

Open Peer Review

PEER REVIEW HISTORY (review-history.pdf). An accounting of the reviewer comments and feedback.

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
