## [Reviewer comments · mSystems]

Dysbiosis of fecal virome in pediatric Crohn's disease and its dynamic changes during infliximab therapy

Ting Ge, Tingting Zhao, Yangming Ruan, Lin Ye, Yongmei Xiao, Fangfei Xiao, Youran Li, Xiaolu Li, Ruixue Wang, Hui Hu, Chunyan Lu, Hong Sun, Chiyu Zhang, Guangjun Yu, and Ting Zhang

Corresponding Author(s): Ting Zhang, Children's Hospital of Shanghai

Review Timeline:

Submission Date:	October 21, 2025
Editorial Decision:	November 14, 2025
Revision Received:	November 21, 2025
Editorial Decision:	December 18, 2025
Revision Received:	December 23, 2025
Accepted:	January 16, 2026

Editor: Hongwei Zhou

Reviewer(s): Disclosure of reviewer identity is with reference to reviewer comments included in decision letter(s). The following individuals involved in review of your submission have agreed to reveal their identity: Qiyi Chen (Reviewer #1); Sahar M. Jawad Abduladheem (Reviewer #3)

Transaction Report:

DOI: <https://doi.org/10.1128/msystems.01489-25>

Re: mSystems01489-25 (**Dysbiosis of fecal virome in pediatric Crohn's disease and its dynamic changes during infliximab therapy**)

Dear Prof. Ting Zhang:

Revision Guidelines

Sincerely,
Hongwei Zhou
Editor
mSystems

Reviewer #1 (Comments for the Author):

The manuscript has made substantial progress since the initial submission, with the authors thoughtfully addressing most of the concerns raised in the first round of reviews.

The added justification for focusing on the virome over the bacteriome is particularly welcome, emphasizing its underexplored role in modulating microbial dynamics, immune responses, and treatment outcomes in pediatric Crohn's disease (CD), which

aligns well with emerging evidence in the field. The clarification on the REPLI-g WTA Single Cell Kit's use for DNA virus amplification, despite its primary RNA targeting, is adequately supported by the provided reference and protocol optimizations, though it still underscores the DNA-centric bias of the study. Similarly, the expanded abstract now better highlights the clinical significance, such as the potential for virome-based biomarkers to improve diagnostic accuracy (AUC=89.3%) and predict infliximab (IFX) response. Updates to figures with explicit sample sizes (e.g., n=53 for the IFX-treated subset, n=25 paired samples) and explanations for absent significance annotations enhance transparency, while the polished language, consistent abbreviations (e.g., distinguishing "post-IFX" from baseline), and added discussions on functional virus-bacteria interactions with new citations improve overall readability and depth.

Despite these improvements, several issues persist. Foremost, the broad use of "virome" throughout the title, abstract, and text is misleading, as the methods explicitly target DNA viruses via VLP enrichment and DNase treatment, excluding RNA viruses which recent studies suggest play roles in IBD inflammation. To avoid overstating the scope, the authors should qualify this as "DNA virome" consistently, perhaps revising the title to "Dysbiosis of fecal DNA virome in pediatric Crohn's disease and its dynamic changes during infliximab therapy," and add a limitation discussing how RNA viruses (e.g., picornaviruses) might complement future investigations through metatranscriptomics.

Second, the small sample size for non-remitters (n=12) and paired longitudinal samples (n=25) continues to limit statistical power, particularly for subgroup comparisons like IFX-R versus IFX-NR in Figure 7. Incorporating a post-hoc power analysis for key outcomes (e.g., virome shifts in remission) would strengthen confidence in the results; alternatively, explicitly state that these findings are preliminary and require validation in larger cohorts.

Thirdly, functional claims, such as the shift from DNA repair to viral replication/assembly, remain somewhat speculative based on GO annotations alone, lacking direct evidence of active phage processes. The author could cite mechanistic previous studies during IFX in IBD models. For example, phage-induced lysis could affect butyrate production in *Blautia*.

Overall, I recommend minor revision. This work will make a meaningful contribution to understanding the virome's role in pediatric CD therapy.

Responses to Reviewer 1

We would like to thank you for your thoughtful and constructive feedback on our manuscript titled “Dysbiosis of fecal virome in pediatric Crohn's disease and its dynamic changes during infliximab therapy”, We have carefully considered your comments and have revised the manuscript accordingly. Below are our responses to your specific concerns:

1. Despite these improvements, several issues persist. Foremost, the broad use of "virome" throughout the title, abstract, and text is misleading, as the methods explicitly target DNA viruses via VLP enrichment and DNase treatment, excluding RNA viruses which recent studies suggest play roles in IBD inflammation. To avoid overstating the scope, the authors should qualify this as "DNA virome" consistently, perhaps revising the title to "Dysbiosis of fecal DNA virome in pediatric Crohn's disease and its dynamic changes during infliximab therapy," and add a limitation discussing how RNA viruses (e.g., picornaviruses) might complement future investigations through metatranscriptomics.

Answer: We apologize for not providing sufficient detail regarding the REPLI-g WTA Single Cell Kit in our previous response.

Total nucleic acids (including DNA and RNA) were extracted using the Magnetic Viral DNA/RNA Kit (TIANGEN), amplified with the REPLI-g WTA Single Cell Kit (QIAGEN) as previously described. Nucleic acids were first reverse-transcribed to complementary DNA (cDNA) using random primers, and then subjected to a ligation reaction to obtain large-fragments of DNA. The obtained DNA was further amplified using MDA with REPLI-g SensiPhi DNA polymerase for 2 h. The amplified products were used for library construction and sequencing. **Thus, this method supports the simultaneous detection of both DNA and RNA viruses and we have added appropriate references (ref. 25) in the methodology section.** In practice, we found that the RNA virus family *Virgaviridae* was less abundant in CD (page 14, line 270), while specific RNA viruses, such as *Pepper mild mottle virus* and *Cucumber green mottle mosaic virus*, were enriched in HC and contributed to our classification model (Figure 5). A recent study that separately extracted DNA and RNA (ref. 33) also reported that “no significant differences were observed between groups at the family and the genus level” for the RNA virome, which aligns with our finding that the most robust dysbiosis signals were within the DNA virome. Therefore, we have clarified this in the revised manuscript (page 9, lines 166) and given that our methodology allowed for the detection and analysis of some RNA viral signals, we believe that retaining the broader term “virome” in the title best reflects the analytical scope and findings of our study. We appreciate your suggestion and will devote more attention to the relationship between RNA viruses and IBD in our future research through metatranscriptomics.

2. Second, the small sample size for non-remitters (n=12) and paired longitudinal samples (n=25) continues to limit statistical power, particularly for subgroup comparisons like IFX-R versus IFX-NR in Figure 7. Incorporating a post-hoc power analysis for key outcomes (e.g., virome shifts in remission) would strengthen confidence in the results; alternatively, explicitly state that these findings are preliminary and require validation in larger cohorts.

Answer: We appreciate your insightful concern about the small sample size and its impact on statistical power. We have acknowledged this limitation in the manuscript (page 23, lines 468-471)

and stated that these findings are preliminary and require validation in larger cohorts to confirm the observed trends (page 23, lines 465~467) in the revised manuscript.

3. Thirdly, functional claims, such as the shift from DNA repair to viral replication/assembly, remain somewhat speculative based on GO annotations alone, lacking direct evidence of active phage processes. The author could cite mechanistic previous studies during IFX in IBD models. For example, phage-induced lysis could affect butyrate production in Blautia.

Answer: We appreciate your feedback on this point. In response, we have carefully revised the manuscript to present the findings with greater caution. Specifically, we have adjusted the language to clarify that the observed shift in functions from DNA repair to viral replication/assembly is based on GO annotations, and we acknowledge that further research, including mechanistic studies and experimental validation, is necessary to substantiate these functional shifts. To strengthen the context, we have cited recent literature (ref. 42), which explores phage-mediated processes in IBD models and provides a more robust framework for understanding the potential mechanisms involved in these shifts.

Thank you once again for your valuable feedback. Should you have any further concerns, please do not hesitate to let us know, and we will be happy to address them.

Re: mSystems01489-25R1 (**Dysbiosis of fecal virome in pediatric Crohn's disease and its dynamic changes during infliximab therapy**)

Dear Prof. Ting Zhang:

Revision Guidelines

Sincerely,
Hongwei Zhou
Editor
mSystems

Reviewer #1 (Comments for the Author):

the authors have properly addressed the reviewer's concern

Reviewer #3 (Comments for the Author):

Study Weaknesses

- Limited sample size, especially in the non-remission group (n = 12), reducing statistical power.
- Only one post-treatment timepoint was collected, limiting the ability to evaluate dynamic virome changes over the full treatment course.
- Sampling restricted to fecal material, which may not fully represent mucosal virome communities implicated in Crohn's disease.
- Geographic and single-center cohort, limiting generalizability to broader pediatric populations.
- Potential confounding factors (diet, environment, undetected viral exposures) not fully controlled.
- Lack of mechanistic experiments to validate how specific viral taxa influence inflammation or treatment response.
- Cross-sectional differences dominate, making it difficult to establish causality between virome shifts and disease activity.

Study Weaknesses

- **Limited sample size**, especially in the non-remission group (n = 12), reducing statistical power.
- **Only one post-treatment timepoint** was collected, limiting the ability to evaluate dynamic virome changes over the full treatment course.
- **Sampling restricted to fecal material**, which may not fully represent mucosal virome communities implicated in Crohn's disease.
- **Geographic and single-center cohort**, limiting generalizability to broader pediatric populations.
- **Potential confounding factors** (diet, environment, undetected viral exposures) not fully controlled.
- **Lack of mechanistic experiments** to validate how specific viral taxa influence inflammation or treatment response.
- **Cross-sectional differences dominate**, making it difficult to establish causality between virome shifts and disease activity.

Responses to Reviewer 3

We sincerely thank for your thoughtful and constructive feedback on our manuscript. We have carefully considered all points and revised the manuscript accordingly. Our point-by-point responses are detailed below.

1. Limited sample size, especially in the non-remission group (n=12), reducing statistical power.

Answer: We greatly appreciate your insight on this matter, as the sample size is indeed a critical factor when interpreting our findings. As discussed on page 23 (Lines 460-462), we have explicitly acknowledged the limitation of the small sample size and its potential impact on statistical power in the revised manuscript. Conclusions related to virome signatures predictive of IFX response are framed as preliminary and hypothesis-generating, emphasizing the need for future validation in larger, multi-center cohorts.

2. Only one post-treatment timepoint was collected, limiting the ability to evaluate dynamic virome changes over the full treatment course.

Answer: We fully acknowledge that multiple timepoints would provide a more comprehensive view and are grateful for this suggestion. We hope to explore this in future studies. The single post-treatment timepoint (after the third IFX infusion) was selected to capture early virome response to induction therapy. We have acknowledged this limitation in the discussion (Page 23, Lines 463-467), stating that future longitudinal studies with repeated sampling throughout the maintenance phase are needed to fully understand the temporal evolution of the virome in response to IFX (Page 23, Lines 475-477).

3. Sampling restricted to fecal material, which may not fully represent mucosal virome communities implicated in Crohn's disease.

Answer: Thank you for this valid point. We recognize that mucosal virome may be more directly relevant to intestinal inflammation. A sentence has been added in the discussion (Page 23, Lines 467-469) to acknowledge this limitation and suggest that future research incorporating mucosal biopsies would complement our findings (Page 23, Lines 475-477).

4. Geographic and single-center cohort, limiting generalizability to broader pediatric populations.

Answer: We thank the reviewer for this comment. Our cohort was indeed recruited from a single center in Shanghai, China. Geographic and ethnic factors are known to influence microbiome composition. We have now clearly stated this as a limitation in the revised manuscript (Page 23, Lines 469-471) and have emphasized the importance of multi-center studies involving diverse populations to validate and extend the generalizability of our results (Page 23, Lines 475-477).

5. Potential confounding factors (diet, environment, undetected viral exposures) not fully controlled.

Answer: We truly appreciate your suggestion to consider factors like diet and environment more

rigorously. While we did collect some basic metadata, we agree that future studies should account for these variables in greater depth. We have added a statement in the discussion (Page 23, Lines 469-471) acknowledging that unmeasured confounding factors could influence our results and that future studies should aim to control for these variables more rigorously.

6. Lack of mechanistic experiments to validate how specific viral taxa influence inflammation or treatment response.

Answer: We appreciate this insightful comment. Our study is primarily observational and descriptive. We have now explicitly stated in the discussion (Page 23, Lines 472-473) that the functional roles of the identified viral taxa are speculative based on correlations and that future work using gnotobiotic animal models or in vitro systems is necessary to establish causal mechanisms (Page 23, Lines 475-477). We hope our findings will serve as a foundation for the mechanistic studies suggested by the reviewer.

7. Cross-sectional differences dominate, making it difficult to establish causality between virome shifts and disease activity.

Answer: We thank the reviewer for this insightful comment. We have clarified in the manuscript that while the baseline comparison between CD and HC is cross-sectional, the analysis of IFX response incorporates a longitudinal element (comparing pre- and post-treatment samples within the same individuals). Nevertheless, we acknowledge that establishing definitive causality from human association studies remains challenging. We have revised the discussion (Page 23, Lines 458-459) to reflect this limitation more precisely, cautioning against causal interpretations and framing our findings as identifying associations.

All acknowledgments of limitations suggested above have been incorporated into the discussion section of the revised manuscript. We believe these revisions provide a more balanced and critical interpretation of our data. We hope the responses are satisfactory.

Thank you again for the constructive feedback.

Re: mSystems01489-25R2 (**Dysbiosis of fecal virome in pediatric Crohn's disease and its dynamic changes during infliximab therapy**)

Dear Prof. Ting Zhang:

Your manuscript has been accepted, and I am forwarding it to the ASM production staff for publication. Your paper will first be checked to make sure all elements meet the technical requirements. ASM staff will contact you if anything needs to be revised before copyediting and production can begin. Otherwise, you will be notified when your proofs are ready to be viewed.

Sincerely,
Hongwei Zhou
Editor
mSystems